# Fabrication of Thymoquinone and Ascorbic Acid-Loaded Spanlastics Gel for Hyperpigmentation: In Vitro Release, Cytotoxicity, and Skin Permeation Studies

**DOI:** 10.3390/pharmaceutics17010048

**Published:** 2025-01-02

**Authors:** Ahlam Zaid Alkilani, Rua’a Alkhaldi, Haneen A. Basheer, Bassam I. Amro, Maram A. Alhusban

**Affiliations:** 1Department of Pharmacy, Faculty of Pharmacy, Zarqa University, Zarqa 13110, Jordan; 20209212@zu.edu.jo (R.A.); hbasheer@zu.edu.jo (H.A.B.); malhussban@zu.edu.jo (M.A.A.); 2Department of Pharmaceutics and Pharmaceutical Technology, School of Pharmacy, The University of Jordan, Amman 11941, Jordan; amrob@ju.edu.jo

**Keywords:** dermal, spanlastics, thymoquinone, ascorbic acid, hyperpigmentation, nanomedicine

## Abstract

**Background/Objectives**: The demand for a safe compound for hyperpigmentation is continuously increasing. Bioactive compounds such as thymoquinone (TQ) and ascorbic acid (AA) induce inhibition of melanogenesis with a high safety profile. The aim of this study was to design and evaluate spanlastics gel loaded with bioactive agents, TQ and AA, for the management of hyperpigmentation. **Methods**: Several spanlastics formulations were successfully fabricated and characterized in terms of morphology, vesicle size, zeta potential, and release. **Results**: The optimized TQ-loaded spanlastic formulation showed an average size of 223.40 ± 3.50 nm, and 133.00 ± 2.80 nm for AA-loaded spanlastic formulation. The optimized spanlastics formulation showed the highest entrapment efficiency (EE%) of 97.18 ± 2.02% and 93.08 ± 1.95%, for TQ and AA, respectively. Additionally, the edge activator concentration had a significant effect (*p* < 0.05) on EE%; it was found that by increasing the amount of EA, the EE% increases. Following that, the optimal spanlastics fomulation loaded with TQ and AA were incorporated into gel and explored for appearance, pH, spreadability, stability, rheology, in vitro release, ex vivo permeation study, and MTT cytotoxicity. The formulated spanlastics gel (R-1) has a pH of 5.53. Additionally, R-1 gel was significantly (*p* < 0.05) more spreadable than control gel, and exhibited a shear thinning behavior. Most importantly, ex vivo skin deposition studies confirmed superior skin deposition of TQ and AA from spanlastic gels. Additionally, results indicated that tyrosinase inhibition was primarily due to TQ. When comparing TQ alone with the TQ-AA combination, inhibition ranged from 18.35 to 42.73% and 24.28 to 42.53%, respectively. Both TQ spanlastics and the TQ-AA combination showed a concentration-dependent inhibition of tyrosinase. **Conclusions**: Spanlastic gel might represent a promising carrier for the dermal delivery of TQ and AA for the management of hyperpigmentation conditions.

## 1. Introduction

Dermatologists are seeking long-term topical skincare solutions to address issues related to skin hyperpigmentation such as melasma and freckles [1]. Additionally, many women try to lighten their skin tone using whitening cosmetics products. Even though various synthetic compounds are currently available in the world market, human toxicity is increasing with traditional lightening agents like hydroquinone which are highly effective, but they have a poor safety profile including ochronosis, carcinogenic potential, and other local or systemic side effects when used over a prolonged period of time [2,3]. The demand for a safe compound for hyperpigmentation is continuously increasing. Exploring the benefits of naturally occurring ingredients opens the door to developing safer alternatives for treating skin hyperpigmentation disorders [4]. Bioactive compounds such as thymoquinone, antioxidants, and polyphenols induce inhibition of melanogenesis through various mechanisms with a high safety profile [4,5]. Three melanocyte-specific enzymes, tyrosinase, tyrosinase-related protein (TRP)-1, and TRP-2, are involved in melanogenesis [6]. Most international regulations have prohibited the use of some synthetic lightening agents [3,7]. In spite of current regulations, lightening agents still dominate the market [8,9,10]. Consequently, the transition to more natural and bioactive compounds has become a demand.

Thymoquinone (TQ) is a natural chemical derived from the black seeds of Nigella sativa, proven to have anti-cancer and anti-inflammatory properties. TQ is a hydrophobic compound and the most active constituent of the volatile oil of Nigella sativa seeds, “black seeds”. They have a very low degree of toxicity [11,12]. TQ exerts its effect by acting as a free radical and superoxide radical scavenger, meanwhile preserving the activity of various anti-oxidant enzymes such as catalase, glutathione peroxidase, and glutathione-S-transferase [13]. TQ shows a decrease in tyrosinase activity and tyrosinase expression, meaning it acts as a lightening agent [14,15]. In spite of having a numerous beneficial effect on the skin, pharmaceutical utilization of TQ has been limited due to challenges during fabrication like high hydrophobicity (log *p* = 2.54) [16], very low aqueous solubility (<1.0 mg/mL) [17], and chemical instability [18,19,20,21]. TQ is significantly affected by light and heat, leading to its degradation. One of the primary degradation products of TQ is its dimer, which forms under the presence of light [17,22].

Ascorbic acid (AA) is a potent antioxidant that removes reactive oxygen species (ROS). It is also known to affect multiple stages in collagen production [23,24,25,26]. Many derivatives of vitamin C are available in the market, such as ascorbate phosphate, ascorbyl palmitate, and ascorbic acid sulfate, which have higher stability than ascorbic acid but low activity and efficacy compared with AA [27,28]. The bioavailability of AA in the skin is insufficient when administered orally [29,30,31]. As a result, oral administration of ascorbic acid to peripheral tissues such as the skin is not enough for pharmaceutical application.

Dermal drug delivery is one of the most attractive methods to deliver bioactive compounds through healthy skin [32]. Compared to other routes of administration, the dermal drug delivery system provides several benefits, including a painless method and a non-invasive method that can serve as a substitute for the parenteral route and avoid first-pass metabolism [32,33]. Additionally, dermal delivery has a better pharmacokinetic profile, which reduces the possibility of undesirable side effects and controlled release of drugs [34,35]. This study addresses hyperpigmentation as clarified in the title, focusing on delivering TQ and AA to the skin layers where melanogenesis occurs. Melanocytes are present in the epidermis and hair follicles [36,37]. Therefore, the purpose of the study is to achieve dermal delivery that can effectively reduce melanin production. This approach avoids systemic side effects, offering a promising strategy for treating hyperpigmentation in a controlled manner. However, the skin’s strong barrier function presents a significant challenge to the effectiveness of topical formulations, making drug penetration through the skin difficult [38]. Therefore, the encapsulation of TQ and AA in nanoformulation can improve drug penetration and the stability of drugs [39,40]. Spanlastics are nanovesicles characterized by high elasticity and deformability [41,42]. Because of their flexibility, they are able to pass through the stratum corneum by squeezing them through skin pores [43]. Spanlastics are vesicle-based drug delivery systems. They contain only non-ionic surfactants and edge activators [41]. In comparison with niosome, spanlastics showed a higher EE%, better release profile, and higher permeation through the biological membrane [43,44].

The biggest challenge in formulating AA and TQ is maintaining their stability and improving their penetration to the skin since they are photosensitive compounds, and AA is hydrophilic. To overcome their stability issue and enhance drug permeation, this study aimed to use spanlastics as a drug carrier for the encapsulation and delivery of dual compounds, TQ and AA, to the skin, forming a nanocarrier-based system and evaluating spanlastics gel formulation in managing hyperpigmentation. Initially, spanlastics were loaded with TQ and AA using the ethanol injection method, followed by the fabrication and characterization of a gel formulation incorporated with nanovesicles. After that, the cytotoxicity of the spanlastics gel was investigated using the MTT assay, and finally, the tyrosinase inhibition activity of TQ and AA was studied for pharmaceutical applications.

## 2. Materials and Methods

SEPIGEL 305™ and Benzyl Alcohol was gifted from Ozone Cosmetic & Natural Products Mfg., Co. (Amman, Jordan). Thymoquinone (TQ) was purchased from Sigma Aldrich™ (Dorset, UK). HPLC-grade acetonitrile, methanol, ethanol, 2-propanolol, and HPLC-grade water were purchased from Tedia™ (Fairfield, OH, USA). Acetic acid glacial and ascorbic acid (AA) was purchased from MEDEX™ (Mallow, Ireland). Span 60, and Tween 20, sucrose and phosphate-buffered saline (PBS) tablet was purchased from Sigma Aldrich™ (Dorset, UK). All other chemicals used in this work are of analytical grade.

### 2.1. Method of Analysis

The analysis of TQ and AA was conducted using reverse-phase high-performance liquid chromatography (RP-HPLC). A C8 column (250 × 4.6 mm; 5 μm, Zorbax, Agilent, San Diego, CA, USA) was used with a mobile phase composed of 1 *v*/*v*% acetic acid (pH = 3.5) and acetonitrile in a ratio of 60:40 *v*/*v*%. The analysis was performed at a flow rate of 0.5 mL/min and an oven temperature of 25 °C. The HPLC system was Shimadzu LC-20AT Pump, Standard Autosampler, SPD-20A UV/VIS Detector (Shimadzu, Kyoto, Japan). All injections for HPLC were 20 μL, and standard solutions for TQ and AA were prepared by diluting stock solutions, resulting in final concentrations of 4 to 18 μg/mL for TQ and 40 to 180 μg/mL for AA. The detection of the drugs was performed at λmax 254 nm, with retention times of 2.5 min for AA and 10.2 min for TQ.

### 2.2. Equilibrium Solubility of TQ

Equilibrium solubility of TQ at 37 ± 1 °C was determined in the following solvents: PBS alone (pH = 7.4), PBS containing 40% polyethylene glycol 400 (PEG 400), and PBS containing 40% ethanol. In an amber glass vial, an excess amount of the TQ was added to 2 mL of solvent. The vial was placed in a shaker incubator at 37 ± 1 °C and at shaking speed of 100 rpm for 24 h. Aliquots of the solution were withdrawn and centrifuged at 1400 rpm for 15 min, then suitably diluted and the concentration of TQ measured. The concentration of TQ was determined by HPLC analysis. All the experiments were repeated in triplicate and the results were presented as mean ± SD.

### 2.3. Formulation of Spanlastics

Spanlastics of TQ were formulated by using an ethanol injection technique as shown in Figure 1. A weighed amount of Span 60 was dissolved in 10 mL of ethanol. After that, a known amount of TQ was added and mixed well. The final solution was then gradually and slowly injected into 10 mL of 9% sucrose solution (pH: 7.4) which contained the edge activator (EA) by using a syringe pump at a rate of 1 mL/min. The solution was then constantly stirred with a magnetic stirrer at 800 rpm while the temperature was kept at 60–65 °C. Spanlastics were sonicated for 5 min and then kept at 4 °C in a refrigerator for 24 h for further characterization.

The ethanol approach as previously described was also used to fabricate AA spanlastics as shown in Table 1. Briefly, Span 60 and 100 mg of AA were dissolved in 20 mL ethanol as an organic phase. The organic solvent was then injected into 20 mL of 9% sucrose solution (pH: 7.4) which contained EA by using a syringe pump at a flow rate of 1 mL/min. The solution was stirred with a magnetic stirrer at 1200 rpm and the temperature was kept at 60–65 °C. For further characterization, spanlastics were sonicated for 5 min and then kept at 4 °C in a refrigerator for 24 h.

### 2.4. Characterization of Spanlastics

#### 2.4.1. Transmission Electron Microscope (TEM)

The morphology of spanlastics formulation was studied using a transmission electron microscope (TEM), operating voltage of 60 kV FEI Morgani 268™, (Eindhoven, The Netherlands) coupled to a Mega View II digital camera. Spanlastics suspension was diluted with distilled water and then put onto a carbon-coated copper grid. ImageJ software Version 1.54 was used to show the morphology of spanlastics.

#### 2.4.2. Particle Size (PS) and Polydispersity Index (PDI)

A particle size analyzer (Brookhaven NanoBrook 90Plus Particle Size Analyzer, Holtsville, NY, USA) was used to assess the particle size of spanlastics by utilizing a dynamic light scattering technique. A total of 100 µL of the prepared spanlastics suspension were diluted up to 2000 µL using distilled water, then sonicated for 10 min until a clear solution was obtained, and analyzed using particle size analyzer software (v4). Sonication was used prior to particle size and zeta potential analysis to ensure the dispersion of any aggregated vesicles and obtain accurate measurements. This step helps break down loosely associated aggregates that may form during storage or preparation. Also, it was used only for a short duration (10 min). The findings of each experiment were provided as mean ± SD, and each experiment was carried out in triplicate.

#### 2.4.3. Zeta Potential (ZP)

The ZP of spanlastics formula was evaluated using Zetasizer (Brookhaven NanoBrook 90Plus Particle Size Analyzer, Holtsville, NY, USA). The surface charges of the particles were determined via electrophoretic light scattering (ELS). At 25 °C, 100 µL of the prepared spanlastics suspension was diluted up to 2000 µL distilled water and then sonicated for 10 min until a clear solution was obtained. The results of each experiment were performed in triplicate and reported as mean ± SD.

#### 2.4.4. Entrapment Efficiency (EE%)

Separation of TQ and AA-loaded spanlastics was done by the ultracentrifugation method, where 1 mL of each formula was centrifuged at 12,000 rpm for 30 min at 4 °C using a refrigerated centrifuge (Hermele Universal Centrifuge Z326, Gosheim, Germany) [35]. The clear supernatant was separated and discarded to obtain the entrapped spanlastics and the process was repeated twice to ensure the complete removal of free drugs. The resultant spanlastics were lysed with n-propanol and sonicated for 5 min. The EE% of each formula was calculated using Equation (1).
(1)$$EE\left(\%\right)=\frac{Entrapped\;amount}{Total\;amount}\times 100\%$$

#### 2.4.5. Attenuated Total Reflection-Fourier Transform Infrared Spectroscopy (ATR-FTIR)

Attenuated Total Reflection-Fourier Transform Infrared spectroscopy (ATR-FTIR) was performed for Tween 20, Span 60, TQ, AA, physical mixtures of each formula, and spanlastics. The spectra were obtained in the wave number range of 4000–400 cm^−1^ at 2 cm^−1^ resolutions and with 32 scans per spectrum. ATR-FTIR spectra were analyzed using Ira^®^ FTIR data explorer V 1.0 software [45].

#### 2.4.6. In Vitro Release Study

The in vitro release profile of spanlastics of TQ and AA has been evaluated using the dialysis bag methods, using a dialysis bag (MWCO 12–14 KDa) that had been pre-soaked in PBS (pH 7.4) overnight. After removing the un-entrapped drug, the spanlastics of TQ and AA were reconstituted with 1 mL PBS (pH 7.4) and then put in a cellulose dialysis bag. The dialysis bag was then placed inside a 100 mL flask in a shaker incubator set to 37 °C and 200 rpm for 4 h. For TQ, the release media was 60 mL PBS containing 40% ethanol. Samples of 10 mL from the bottle were withdrawn at 15, 30, 60, 120, 180, and 240 min and immediately replaced with fresh solvent to maintain the sink condition during the experiment. For AA, the procedures were similar, but the volume of media was 55 mL and the withdrawn samples were 5 mL at each point. Both were analyzed using HPLC-UV, and each experiment was carried out in triplicate.

### 2.5. Fabrication of Spanlastics Gels Loaded with TQ and AA

SEPIGEL 305 polymer was chosen as a gelling polymer at a concentration of 2 *w*/*w*%. To prepare spanlastics gels, the selected formula (Q-4 and Z-1) was mixed with a 0.5 g of benzyl alcohol (1% *w*/*w*) and a gel matrix to get the final weight of spanlastics gel 50 g, using a high-speed mixer at 1500 rpm for 2 min Hauschild SpeedMixer™ (Hamm, Germany). The resultant spanlastics gels contained 0.74 mg TQ and 1.9 mg AA per 1 g of gel base. In a similar method, conventional control gels were formulated by the addition of 37 mg TQ and 95 mg AA to the gel base. Once the active ingredients were thoroughly mixed, 0.5 g of benzyl alcohol (1% *w*/*w*) was added. For further characterization, conventional gels were mixed for 2 min and then kept at 4 °C in a refrigerator.

### 2.6. Characterization of Spanlastics Gels

#### 2.6.1. The Appearance and pH of Gel

The appearance of spanlastics gel formulation was compared to conventional gel (control) in terms of gel clarity and color. The pH of spanlastics gel was measured using an Inolab pH meter-720-WTW (Weilheim, Germany). The pH of the gel was directly checked by the direct insertion of the electrode into the sample after preparation.

#### 2.6.2. Spreadability of Gel

The spreadability of the gel was evaluated by applying 0.5 g of the gel on a glass plate and then using a second glass plate. Five minutes was given for a half kilogram of weight to rest on the upper glass plate for both spanlastics gels and control gels [46]. The diameter of the circle after spreading of the gel was determined.

#### 2.6.3. Determination of Drug Content in Gel

Ten grams of spanlastics gels were dissolved in a total of 100 mL solution containing 20:80 distilled water:n-propanol and stirred for 3 h. A specific volume was filtered, and the amount of TQ and AA was determined using HPLC-UV.

#### 2.6.4. Rheology of Gel

The rheological studies (viscosity and viscoelastic measurements) of R-1 gel (R-1) and (C-1) were carried out in triplicate at 32 °C using a controlled-stress rheometer (CSR) (Anton Paar, MCR 302; Graz, Austria) with a cone-plate geometry (gap of 0.1 mm, cone diameter of 25 mm, and cone angle of 1°).

#### 2.6.5. Viscosity Measurements

Before testing, 0.5 g of gel samples were placed onto the plate and allowed to relax for about a minute. Each gel’s viscosity was measured using an applied shear rate that ranged between 0.1 and 100 s^−1^. Gel flow curves were obtained and evaluated [35].

#### 2.6.6. Viscoelastic Measurements

The strain-sweep measurements for all gels were performed to determine the linear viscoelastic (LVE) region. In this region, the elastic (storage modulus (G′)) and viscous (loss modulus (G″)) moduli of gels must remain constant and independent of strain. The frequency-sweep measurements of gels must be performed within the LVE region. The first step was to determine the LVE region, therefore, 0.5 g gel samples were loaded onto the lower plate and left to equilibrate at 32 °C for 1 min, then the upper cone was lowered until the gap between the cone and plate was 0.1 mm. The cone was oscillated at a frequency of 6.28 rad/s during measurement. The G′ and G″ moduli were probed as a function of oscillatory strain ranging between 0.01 and 100. The frequency-sweep measurement was conducted to determine the viscoelastic behavior (G′ and G″) of the gels based on the strain-sweep measurement. Gels were exposed to dynamic oscillatory over a frequency range of 0.1–100 rad/s and at fixed strain selected from the previously determined LVE region.

### 2.7. In Vitro Release Study

To evaluate the release of AA and TQ from spanlastics and conventional gels, a dialysis bag (MWCO 12–14 KDa) was used. A quantity of spanlastics gels was placed in a cellulose dialysis bag. The release media was 60 mL PBS containing 40% *w*/*v* ethanol. Samples of 10 mL from the bottle were withdrawn at 15, 30, 60, 120, 180, and 240 min and immediately replaced with fresh solvent to maintain the sink condition during the experiment. For control gels, the procedures were similar. Both were analyzed using HPLC-UV, and each experiment was carried out in triplicate.

### 2.8. Drug Permeation Study

Ex vivo permeation studies were carried out using Franz diffusion cells (PremeGear, Hellertown, PA, USA) with an aperture diameter of 15 mm and a diffusion surface area of 1.76 cm^2^ to evaluate the permeation of TQ and AA from spanlastics gels. Full-thickness rat skin was taken from the back of a rat. The protocols of rat skin preparation were approved by the Ethics Committee for Scientific Research; approval number (4/2335/27). The rat skin was prepared and clamped between the donor and receptor compartments. The stratum corneum (SC) side was connected to the donor compartment. After that, 1 g of spanlastics gel was placed onto the rat skin. The receiver chamber, which contains 12 mL PBS containing 40% ethanol, was securely fastened using a clamp with the donor compartment. To prevent evaporation, the area between the two chambers was covered with waterproof film (ParafilmTM). The temperature was kept at 32 °C which is similar to the skin surface. Following that, an 8 mL sample was taken from the receptor cell at different intervals of 0.25, 0.5, 1, 2, 3, 4, and 5 h. After each collection, the same volume of receiver solution was used to replace the samples. The samples were withdrawn and analyzed using HPLC.

Subsequently, rat skin was taken, cleaned with PBS to remove any residual TQ and AA, and then homogenized with n-propanolol and stirred for 5 h in the hotplate without heating to extract TQ and AA that remained in the skin. After that, the skin was centrifuged at 12,000 rpm for 20 min and TQ and AA content in the supernatant was quantified by HPLC analysis.

### 2.9. Cell Culture

The Human Dermal Fibroblast (HDF) cell line used in this study was kindly provided by Dr. Walhan Al-Sha’er from the Stem Cells Therapy Center at the University of Jordan (Amman, Jordan). The HDF cells were cultured and maintained in a humidified incubator at 37 °C with 5% CO₂. The growth medium was Dulbecco’s Modified Eagle Medium (DMEM) High Glucose, supplemented with 10% fetal bovine serum (FBS), 1% 2 mM L-glutamine, 1% MEM Non-Essential Amino Acid (100×), 1% 100 mM sodium pyruvate, and 1% penicillin (10,000 U/mL)–streptomycin (10 mg/mL).

#### 2.9.1. MTT Assay

The cytotoxicity of R-1 gels and its corresponding blank gel on HDF cells was assessed using the MTT assay. HDF cells were seeded in a 96-well plate at a concentration of 5 × 10^3^ cells/mL by adding 180 μL of cell suspension in DMEM high glucose medium to each well. Cells were allowed to adhere and grow for 24 h in a humidified incubator set to 37 °C with 5% CO₂. Following the incubation period, 20 μL of R-1 gel at various concentrations (1/3.75 mM, 100/375 µM, 50/187.5 µM, 25/93.75 µM, 12.5/46.87 µM, and 6.25/23.43 µM) or the corresponding blank gel concentrations was added to each well, reaching a total volume of 200 μL per well. Additionally, doxorubicin at a concentration of 25 µM was used as a positive control for cytotoxicity. After 72 h of treatment, cell viability was assessed by adding 20 μL of MTT stock solution (5 mg/mL; Sigma, St. Louis, MO, USA) to each well, followed by a 4-h incubation at 37 °C to allow for the formation of formazan crystals. Upon completion of the incubation, the MTT-containing medium was carefully aspirated, and 150 μL of dimethyl sulfoxide (DMSO) was added to each well to dissolve the purple formazan crystals. Absorbance was measured at 570 nm using a colorimetric plate reader. The percentage of viable cells was calculated relative to untreated control cells, thereby enabling the assessment of the cytotoxic effects of various concentrations of R-1 gel and its corresponding blank gel on cell viability. Each experiment was performed in triplicate to ensure reproducibility.

#### 2.9.2. Tyrosinase Inhibitor Assay

The tyrosinase inhibition activity of spanlastic formulations containing TQ and AA was assessed using the Tyrosinase Inhibitor Assay Kit™ (Attogene^®^, Austin, TX, USA). Following the manufacturer’s instructions, a master mix was prepared, and 225 µL aliquots were added to each well of a 96-well plate. Subsequently, 40 µL of each concentration of the spanlastic test samples which are TQ alone, AA alone, and a mixture of TQ and AA was added to the wells, with final concentrations of 100 µM, 50 µM, 25 µM, 12.5 µM, 6.25 µM, and 3.125 µM per well for each. Finally, 7 µL of substrate (L-Tyrosine) was added to each well.

The components were thoroughly mixed by pipetting up and down 3–4 times, and the plate was incubated at room temperature for 5 h. Absorbance readings were then obtained using an Epoch™ Microplate Spectrophotometer (BioTek™, Winooski, VT, USA) at a wavelength of 520 nm. The percentage of tyrosinase inhibition activity was then calculated relative to the control, enabling quantification of inhibitory effects in the presence of the spanlastic formulations.

### 2.10. Short-Term Stability Study

The optimized spanlastics formula (Q-4, Z-1) was assessed after 2 months of the preparation to evaluate PS, PDIs, and EE%. The spanlastics formula was stored in a refrigerator at 4 °C. In addition, short-term stability was carried out on the spanlastics gel (R-1) to investigate the physical and organoleptic qualities like color, appearance, pH, spreadability, and drug content. The spanlastics gel was stored in a refrigerator at 4 °C.

### 2.11. Statistical Analysis

All data are presented as mean ± standard deviation (SD). All experiments were independently performed in triplicate. The mean values of the results were compared using a two-tailed *t*-test in Microsoft Excel from Microsoft 365. *p*-values were calculated to determine the statistical significance of the results, * *p* < 0.05 and ** *p* < 0.01.

## 3. Results and Dicussions

### 3.1. Equilibrium Solubility of TQ

Equilibrium solubility of TQ at 37 ± 1 °C was determined in the following solvents: PBS (pH = 7.4), PBS containing 40% PEG-400, and PBS containing 40% ethanol. There was a significant difference in the results (*p* > 0.05). The solubility of TQ is shown in Table 2. The solubility of PBS alone was 0.537 ± 0.124 mg/mL; for PBS containing 40% PEG-400, it was 2.228 ± 0.459 mg/mL; and for PBS containing 40% ethanol, it was 13.450 ± 0.783 mg/mL. These results are in agreement with previous studies since the problem of TQ was poor aqueous solubility [47,48]. Additionally, our results were in agreement with [49], who evaluated the solubility of TQ in different solvents like PBS, PEG, and ethanol. The results showed that the solubility of TQ was the highest in ethanol, then in PEG, and the lowest solubility was in PBS [49]. Based on our results, PBS containing 40% ethanol was selected as a receptor media for the release study to achieve sink conditions. The sink condition is reached when the equilibrium solubility of drugs in the dissolving medium is at least three times the volume required to produce the drug saturation [50].

### 3.2. Spanlastics Formulation

Several formulations of TQ and AA-loaded spanlastics were successfully prepared using the ethanol injection method (Figure 2).

### 3.3. Characterization of Spanlastics

TQ is a hydrophobic drug, so it is encapsulated in the lipid bilayer, while AA is a hydrophilic compound that is encapsulated in the core of the aqueous compartment or in the outer layer (surface) (Figure 3). Spanlastics are amphiphilic structures that allow encapsulation of both the hydrophobic drug within the lipid bilayer and the hydrophilic drug in the core of the aqueous compartment [51]. The temperature of 60–65 °C was applied for a short duration during spanlastics preparation. Light-sensitive compounds were protected by conducting the process using amber containers and wrapping them with aluminum foils. The stability of both was confirmed by the high drug recovery (TQ: 95.34 ± 1.86%, AA: 99.12 ± 0.62%).

The PS of TQ and AA-loaded spanlastics measured by TEM was 251.5 ± 27.5 nm and 122.3 ± 15.6 nm, respectively, while the PS of TQ and AA-loaded spanlastics measured by DLS was 223 ± 3.5 and 133.0 ± 2.80 as shown in Table 3, respectively. Therefore, the PS of Q-4 and Z-1 measured by TEM was relatively close to those obtained by the particle size analyzer. Therefore, the findings of the particle size analyzer in terms of particle size and uniformity were supported by the TEM analyses. In general, the particle size measured by TEM was usually smaller than the DLS method. However, the particle size measured by TEM could be comparable to the DLS method, as described in previous studies [33].

The mean PS and PDI of TQ and AA-loaded spanlastics are summarized in Table 3 (Mean ± SD). The mean PS of TQ formulations was in the range of 108.10 ± 0.20 nm to 223.40 ± 3.50 and 108.60 ± 4.80 nm to 133.00 ± 2.80 nm for AA. Most of spanlastics formula of TQ and AA was within the most preferred range of topical application for dermatological purposes, which was less than 200 nm. Two types of EA were used in the formulation of spanlastics; Tween 80 and Tween 20. By changing EA from Tween 80 to Tween 20 which has a higher HLB value, there was a significant increase in PS from 108.10 ± 0.20 nm to 223.40 ± 3.50 nm for Q-3 and Q-4, respectively. Therefore, EA type had a negative effect on spanlastic size. Spanlastic vesicles generated with Tween 20 (HLB 16.7) as an EA were bigger than those prepared with Tween 80 (HLB 14.9). This could be attributed to the lower hydrophilicity and lower HLB of Tween 80, resulting in lower surface energy and smaller nanovesicles [52,53]. Our findings were in agreement with a previous study [54] which reported the direct relationship between vesicle size and HLB of edge activator. It was found also that by decreasing the amount of Tween 20 from 200 to 100 mg (Z-1 vs. Z-3), there was a significant decrease (*p* < 0.05) in the spanlastics size which was 133.00 ± 2.80 to 108.60 ± 4.80 nm, respectively for Z-1 and Z-3, thus decreasing the amount of edge activator shows a decrease in PS. This could be attributed to the reduction of surface tension, facilitating particle partition and the formation of smaller vesicles [41,55,56].

In general comparison, the PS of AA-loaded spanlastics had a lower PS than TQ spanlastics. The reason behind this result was the speed of stirring during the preparation of spanlastics, for example, it was 800 rpm for TQ with higher PS (223.40 nm), while it was 1200 rpm for AA with lower PS (133.00 nm) [43].

Zeta potential (ZP) measurements provide the degree of repulsion forces between nanoparticles and the attraction forces between them. ZP is one of the most important criteria for the characterization of nanoparticles due to its effect on nanoparticle stability and in vivo drug delivery performance [41]. The ZP range of TQ spanlastics was from −21.50 ± 1.72 to −14.90 ± 1.91 and −19.5 ± 1.27 to −8.98 ± 0.07 for AA spanlastics. The ZP for Q-4 and Z-1 was −21.50 ± 1.72 and −19.50 ± 1.27, respectively, which is sufficient to provide acceptable repulsion forces between spanlastics, prevent aggregation, and provide stable vesicles [57]. The negative charge value of all formulas even in the absence of a charge inducer came from the free hydroxyl group of surfactant Span 60 molecules [58].

The EE% is the drug loading capacity in nanoparticles which is considered as a key factor in assessing the prepared nanoparticles. Both TQ and AA were successfully entrapped in different formulations. TQ showed a range of EE% between 32.59 ± 1.91 and 97.18 ± 2.02; the highest EE% was for Q-4 spanlastics formulation which used Span 60 and Tween 20 in the ratio of (60:40). The same composition also showed a superior EE% for AA as 93.08 ± 8.56%. Additionally, the edge activator concentration had a significant effect (*p* < 0.05) on EE%; it was found that by increasing the amount of EA, the EE% increases. This was attributed to the effect of EA in forming a layer that increases the vesicle interface stability and gives more space inside spanlastics to hold more drugs [59]. However, the increase in the EE% with the increase in EA concentration up to a certain limit might be attributed to the enhanced fluidity of the vesicle bilayer, which facilitates the leakage of the entrapped drug, resulting in reduced EE% [41,43]. Consequently, the chosen formula Q-4 and Z-1 for gel formulation exhibited high drug entrapment efficiency of 97.18 ± 2.02% and 93.08 ± 1.95%, appropriate zeta potential of −21.50 ± 1.72 and −19.50 ± 1.27 mV, and a vesicle size of 223.40 ± 3.50 nm and 133.00 ± 2.80 nm, which favored their dermal accumulation.

### 3.4. ATR-FTIR Analysis

FTIR Spectra of TQ, Q-4, Blank spanlastics, and physical mix of TQ spanlastics were illustrated in Figure 4A. The main characteristic peaks of TQ were observed in 2800–3000 cm^−1^ due to stretching vibrations of the isopropyl and CH_3_ groups and 1650 cm^−1^ due to stretching of the C = O group [60]. These peaks were disappearing in spanlastics, indicating spanlastics formation. Additionally, it was observed that there were no considerable changes in the characteristic peaks observed in the pure drug spectrum when mixed with excipients, suggesting that the drug was molecularly dispersed in the excipients. Blank spanlastics showed peaks around 3330–3332 cm^−1^ due to the hydroxyl group, with 2926 cm^−1^ indicating carboxylic acid. Tween 20 had a characteristic sharp peak around 3410 cm^−1^ which is attributed to hydrogen-bonded O-H stretching [61]. The bands were shifted to 3332 cm^−1^, and stretched in TQ spanlastics, indicating a strong hydrogen bond between formulation components.

Figure 4B illustrates the FTIR spectra for the formulation of AA in spanlastics (Z-1), blank spanlastics, the physical mixture, and pure AA. The characteristic peak of AA was observed at 1674 cm⁻^1^, corresponding to the C = O group of the lactone ring and broad O-H stretching around 3200–3600 cm⁻^1^ [62]. In the formulation of AA in spanlastics (Z-1), both blank spanlastics and ascorbic acid peaks are present without significant shifts in the characteristic O-H and C = O peaks, suggesting molecular interactions between ascorbic acid and the spanlastics matrix.

FTIR spectra showed that all the characteristic peaks were present and did not exhibit major shifts. Additionally, no new bands were formed, indicating the absence of any considerable chemical interaction and successful encapsulation of AA and TQ within the spanlastics, likely enhancing its stability by protecting it within the matrix.

### 3.5. Fabrication of Gel Loaded with TQ and AA

Q-4 and Z-1 were selected as the optimized formula for further formulation as semisolid formulation. Spanlastics gel (R-1) and conventional control gel (C-1) were fabricated using a SEPIGEL 305™ as a base gel which contains polyacrylamide, C13–14 isoparaffin, and laureth-7. After that, the formulated spanlastics gels were evaluated for their appearance, drug content, pH, spreadability, rheology, in vitro drug release, ex vivo drug permeation, and deposition.

### 3.6. Characterization of Spanlastics Gel

Initially, when the gel base was added to spanlastics formula (Q-4 and Z-1) the color was yellow. After one hour, the color of the gel changed to white as shown in Figure 5, which indicates the drug entrapped inside the polymer as previously reported [63]. The color change from yellow to white indicates successful encapsulation of TQ within the spanlastic gel. Free TQ is yellow, but upon entrapment, its color is masked within the vesicle and gel matrix. High drug recovery percentages further confirm the successful encapsulation. All spanlastics gels were elegant in appearance, homogenous, smooth, shiny, and soft in texture.

The pH of spanlastics gel was found to be 5.53 ± 0.05. Generally, a pH in the range of 4.0 to 7.0 is suitable for topical application [64,65]. These results clearly indicated that spanlastic gel is suitable for use as its pH is close to normal skin pH and will not cause skin irritations [35].

The results of spreadability test for R-1 and C-1 showed that there was a significant difference (*p* > 0.05) in spreadability value for R-1 and C-1 which were 3.65 ± 0.05 and 3.03 ± 0.1, respectively. Therefore, spanlastic gel has good spreadability with low shear as previously described [66].

#### 3.6.1. Determination of Drug Content in Gel

The drug content for TQ and AA in R-1 gel was 95.34 ± 1.86% and 99.12 ± 0.62%, respectively, as shown in Table 4. These results indicate that there was no degradation of the drug in the nanovesicles and the matrices of the gel. The drug content of TQ and AA in the C-1 gel was 54.57 ± 0.49% and 60.72 ± 0.34%, respectively. Our analysis showed a significant difference (*p* < 0.05) in how much TQ and AA were recovered in the R-1 versus C-1 gels. This suggests that TQ and AA are much less stable in their raw form compared to when they are incorporated into nanovesicle formulations [49,62], indicating that the spanlastic formulations enhanced the stability of drugs when encapsulated in nanovesicles.

#### 3.6.2. Rheology of Gel

The viscoelastic nature of gel plays an important role in their adhesion properties, skin spreadability, and retention in the application site.

The viscosity versus shear-rate relation was measured for R-1 and C-1, and the result of the flow curve is shown in Figure 6A. Both R-1 and C-1 exhibit a shear thinning (pseudoplastic fluid) behavior of spanlastics gel, like most gels reported in the literature, which is ideal for topical administration [67]. The shear-thinning behavior is very important for dermal and transdermal gels, which can readily spread over the human skin upon rubbing them [35]. The flow curves showed that R-1 gel displayed a lower viscosity value than their corresponding C-1 gel. The lower viscosity of R-1 might be attributed to the addition of spanlastics to gels, which might interfere with the polymer gel networks, promoting a weakening of the interactions between the inter-polymer connections and hence reducing their thickening efficiency. This is in agreement with [68], who reported that the incorporation of elastic niosomes (spanlastics) into polymer gels markedly decreased their viscosity.

The measuring results of amplitude sweeps for R-1 and C-1 gels are shown in Figure 6B and Figure 6C, respectively. Since it is possible to distinguish between linear and nonlinear viscoelastic behavior using strain amplitude sweep experiments, the modulus of elastic storage (G′) displays the viscous loss modulus (G″) of the soft sample. The amplitude sweep test helps to determine the viscous elastic modulus when the yield stresses fluid in a linear regime. Therefore, the elastic and viscous modulus of the material can be measured by applying oscillatory strain while the frequency is held constant and observing the response of the material [69]. Before crossover, the elastic (G′) was higher than the viscous (G″) modulus since the elastic structure collapses at greater stresses, and the elastic modulus decreases significantly as strain increases. In the crossover strain, G″ becomes bigger than G′ and gels act mostly as liquids for all gel formulations. During the amplitude sweep studies, the LVE zone with low strain values was identified. In this region, gels remained unbroken over the strain range. The yield point (γc) is often known as the end of the LVE area, at which the deformation of gels becomes irreversible, and G’ and G″ change as a function of strain [35,70]. The LVE regions and critical strains of each gel are illustrated in Table 5.

In a frequency test, the frequency of oscillation is ramped with the amplitude held constant. The frequency-sweep measurements were conducted on both gels using a constant strain value of 0.1 percent to determine their viscoelastic characteristics. The measuring results of frequency sweeps for R-1 and C-1 gels are shown in Figure 6D.

In the investigated frequency range (0.1–100 rad/s), the both gels showed a dominance of G′ over G″ (G’ > G″), suggesting stable gels with more solid-like properties [69]. The addition of spanlastics preserved the viscoelastic properties of gel with G’ > G″. Spanlastics were able to strengthen the gel network of R-1, as evidenced by the higher G′ values compared to its conventional gel C-1.

### 3.7. In Vitro Release Study of Spanlastics Gel

In vitro release studies showed a controlled release profile for both Q-4 and Z-1 as shown in Figure 7. These results confirmed the ability of spanlastic vesicles to act as reservoirs for TQ and AA for dermal delivery. This result is supported by the properties of spanlastics; their elasticity allows them to pass through the media [51]. The presence of edge activators can also enhance the fluidity of the vesicle bilayer, facilitating drug release [71]. Additionally, the solubility of drugs in the formulation can also influence the release rate, with more soluble drugs like AA typically releasing more quickly compared with lipophilic drugs like TQ. This behavior is typical for encapsulated systems, where the drug is initially released from the surface, followed by a more gradual and sustained release as the drug diffuses through the vesicle [68].

Figure 7 shows the release profile of the spanlastic gel. After 4 h, the spanlastic gel released 44.76% of TQ and 48.20% of AA which indicates that both compounds are released in a controlled manner from the gel. In the beginning, the immediate release might be due to the free drug in spanlastics gel and the surface-adsorbed drug over vesicles whereas the prolonged release phase after the first hour could be attributed to the release of entrapped AA and TQ from the cores of spanlastics vesicles [72]. All formulations showed a rapid release at the initial stages up to 30 min, followed by a slower, sustained biphasic release profile in spanlastic gel (R-1). Z-1 exhibits a rapid release profile compared to Q-4 and R-1. These findings highlighted the ability of spanlastic vesicles to act as reservoirs for AA and TQ for dermal delivery. Additionally, these results showed that drug release was influenced by the viscoelastic nature of the spanlastics gel which sustained the release of drugs compared with the in vitro release of spanlastics vesicles (Q-4 and Z-1). The incorporation of a vesicular system, spanlastics, within the gel may contribute to the sustained release of TQ and AA which is in agreement with previous studies [52].

### 3.8. Stability Studies

Table 6 shows the results of the short-term stability study for Q-4 and Z-1 at 4 °C. PS, PDI, and EE% were studied as stability parameters. The short-term stability study of Q-4 after 2 months for PS, PDI, and EE% is shown in Table 6. After 2 months, the PS for Q-4 changed from 223.40 ± 3.50 nm to 233.40 ± 2.7 nm, whereas, the PS for Z-1 changed from 133.00 ± 2.80 to 136.6 ± 2.6 nm. These findings suggest that Q-4 and Z-1 spanlastics were stable physically since there were no significant (*p* > 0.05) changes in terms of particle size and PDI. Moreover, evaluating the EE% of the formulation by repeating the experiment after 2 months showed that there were no significant changes (*p* < 0.05) in the EE% value, indicating that TQ-loaded spanlastic and AA-loaded spanlastic were stable during the storage period and there was no leakage. For Q-4, the EE% was 97.18 ± 2.02 and 96.82 ± 1.56 for 0 and 2 months, respectively. For Z-1, the EE% was 93.08 ± 1.95 and 94.66 ± 2.55 for 0 and 2 months, respectively. Short-term stability results showed that the spanlastics formulations were stable at 4 °C in terms of PS, PDI, and EE%.

The spanlastics gel was also evaluated in terms of color, appearance, pH, spreadability, and recovery of TQ and AA after 1 month of storage at 4 °C (Table 7). The findings did not show any significant change in gel clarity and color. Upon visual inspection, no physical changes were observed during storage. Spreadability, color, pH, and recovery were examined after 1 month of storage and showed no difference (*p* > 0.05) when compared to the freshly prepared gel.

### 3.9. Ex Vivo Permeation and Drug Skin Deposition Studies

Ex vivo permeation and drug deposition studies were carried out on full-thickness rat skin using a Franz-diffusion cell. The cumulative permeated amount (Q) after 5 h was 65.49 ± 2.01 μg/cm^2^ and 128.75 ± 0.92 μg/cm^2^ for TQ and AA, respectively, as presented in Table 8. The skin deposition of TQ and AA was 438.05 ± 3.53 μg/cm^2^ and 259.56 ± 5.33 μg/cm^2^, respectively. This result showed a higher accumulation and deposition of the drug in the skin which is better for dermal delivery [73]. These results are in agreement with a previous study [74], which showed that the drug skin deposition of TQ-loaded nanoemulgel was better than the drug-skin-permeated amount of TQ. Additionally, skin deposition results support the ex vivo permeation findings, illustrating that topically applied spanlastic gel could partition across the dermal layer under the influence of the transcutaneous hydration gradient, generating depots from which TQ and AA can be released which is better for the treatment of hyperpigmentation conditions [75].

### 3.10. MTT Cytotoxicity Cell Assay

The cytotoxicity of the R-1 gel and R-1 blank gel was evaluated using the MTT assay on HDF cells, as illustrated in Figure 8. Results indicated that the R-1 blank gel demonstrated a favorable cytotoxicity profile, with cell viability exceeding 84.15 ± 7.04% at the highest concentration used, suggesting minimal cytotoxic effects. These findings are consistent with previous research by Berenguer et al., which reported that SEPIGEL 305 formulations did not exhibit toxicity across various dilutions in multiple cell lines [67]. Additionally, Basheer et al. demonstrated that blank niosomes, at concentrations equivalent to their active formulations, did not show significant cytotoxicity, further validating the safety of inactive components in such formulations [76]. Similarly, Sunoqrot et al. reported that a blank, drug-free nanocapsule formulation was nontoxic against both cancerous and normal cell lines [77].

The results also revealed that R-1 gel achieved a cell viability of 72.36 ± 11.29% at the highest tested concentration, indicating a safe, non-toxic profile. Statistical analysis showed no significant difference between the R-1 blank gel and the control (*p* = 0.06) or between the R-1 gel at the highest concentration and the control (*p* = 0.05). These findings align with recent research which demonstrated that TQ encapsulated in nanoparticles enhanced the drug’s targeting efficiency toward cancer cells while maintaining low toxicity in normal cells [77]. Furthermore, Odeh et al. reported that TQ liposomes exhibited minimal toxicity in normal periodontal ligament fibroblast cells [78]. On the other hand, our data are consistent with previous findings, stating that AA also shows low toxicity when testing AA-loaded NPs or their corresponding Blank NPs even at higher concentrations [26].

### 3.11. Tyrosinase Inhibition Activity for TQ and AA

The tyrosinase inhibition activity of spanlastic formulations containing TQ, AA, and their combination were examined, and data are illustrated in Figure 9. Results indicated that tyrosinase inhibition was primarily due to TQ. When comparing TQ alone with the TQ-AA combination, inhibition ranged from 18.35 to 42.73% and 24.28 to 42.53%, respectively.

Both TQ spanlastics and the TQ-AA combination showed significant concentration-dependent inhibition of tyrosinase (*p* < 0.01), with increased TQ concentrations leading to higher inhibition, consistent with El Khoury et al., who reported that TQ’s inhibitory effect on tyrosinase is concentration-dependent up to a certain point [79].

Although AA is a known lightening agent, it did not exhibit tyrosinase inhibition in this study. This may be due to AA’s alternative mechanisms of action, which do not directly involve tyrosinase. For example, ascorbic acid can reduce skin pigmentation by counteracting ROS induced by UV exposure, which otherwise enhances tyrosinase activity [80]. AA can also prevent melanin formation by reducing o-dopaquinone back to dopa [81]. However, as this experiment used L-tyrosine instead of o-dopaquinone as a substrate, AA’s effect on tyrosinase was not observed here.

## 4. Conclusions

The stability of free AA is a significant challenge, as these compounds are prone to degradation under heat and oxidation in aqueous media. Additionally, the poor aqueous solubility and chemical instability of TQ are also well documented. Therefore, we focused on developing and evaluating the novel nanovesicle-loaded gel system as a means to enhance the stability and efficacy of TQ and AA for dermal delivery. Encapsulation within nanovesicles enhances the stability of these compounds by providing a protective matrix, shielding them from external degradative factors. Notably, the encapsulation efficiency of the formulations containing Tween 20 was significantly higher than that of the formulations prepared using Tween 80. Our findings demonstrated improved drug recovery rates and prolonged release profiles for TQ and AA when encapsulated in spanlastic nanovesicles and incorporated into gel formulations. This will ensure greater bioavailability and efficacy compared to their free forms. To summarize, spanlastic vesicles may be a promising nanocarrier that enhances and controls the dermal delivery of TQ and AA for the management of hyperpigmentation and improves its chemical instability. However, the main limitation of this study includes the lack of in vivo studies, which are essential for further validating the results observed in this research. Therefore, future research could build upon this work to explore additional comparative analyses such as preclinical and clinical studies to assess the safety, efficacy, and potential therapeutic benefits of the developed formulation.

## Figures and Tables

**Figure 1 pharmaceutics-17-00048-f001:**
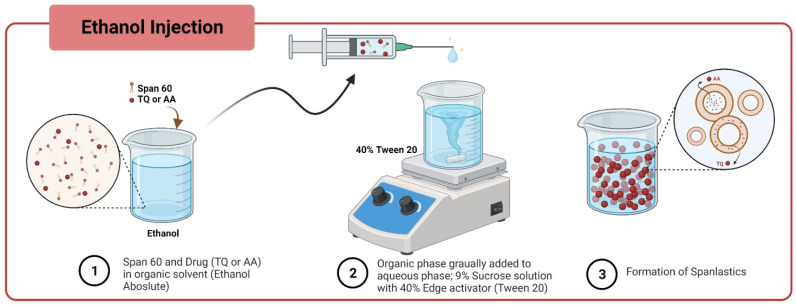
Preparation of TQ and AA Spanlastics using the ethanol injection method.

**Figure 2 pharmaceutics-17-00048-f002:**
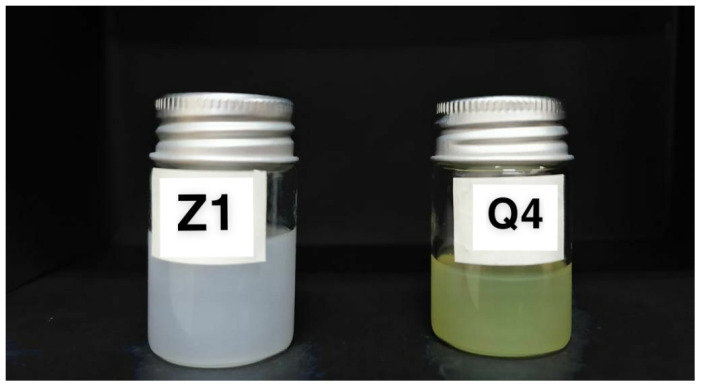
The developed spanlastics formula Z1 and Q4 containing AA and TQ, respectively.

**Figure 3 pharmaceutics-17-00048-f003:**
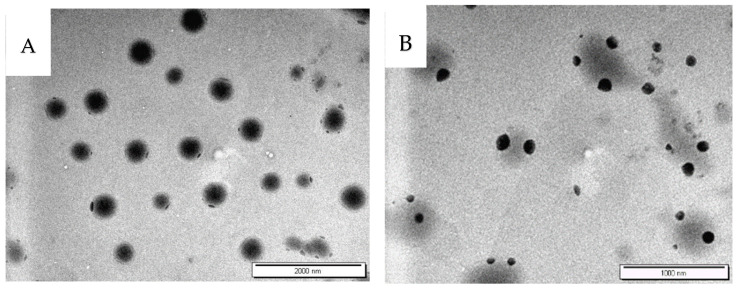
TEM images of spanlastics (**A**) Q-4 and (**B**) Z-1.

**Figure 4 pharmaceutics-17-00048-f004:**
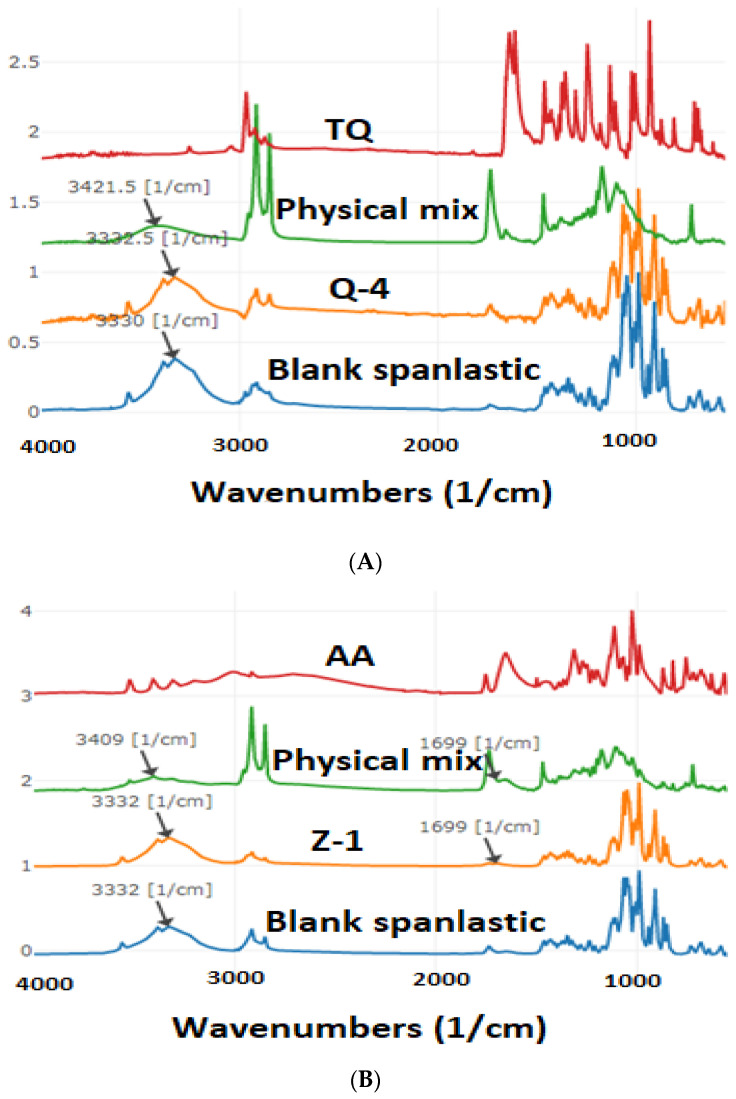
FTIR Spectra of (**A**) Blank spanlastics, TQ, Q-4, and Physical mix and FTIR Spectra of (**B**) Blank spanlastics, AA, Z-1, and Physical mix.

**Figure 5 pharmaceutics-17-00048-f005:**
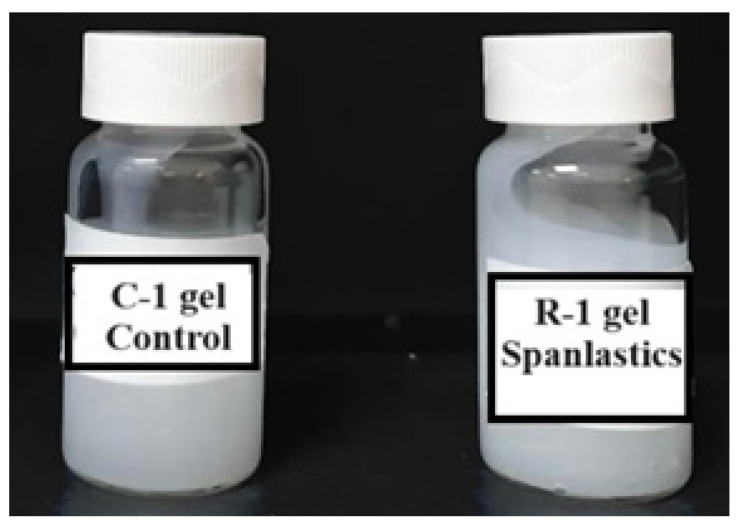
The prepared spanlastics R-1 gel and control gel (C-1).

**Figure 6 pharmaceutics-17-00048-f006:**
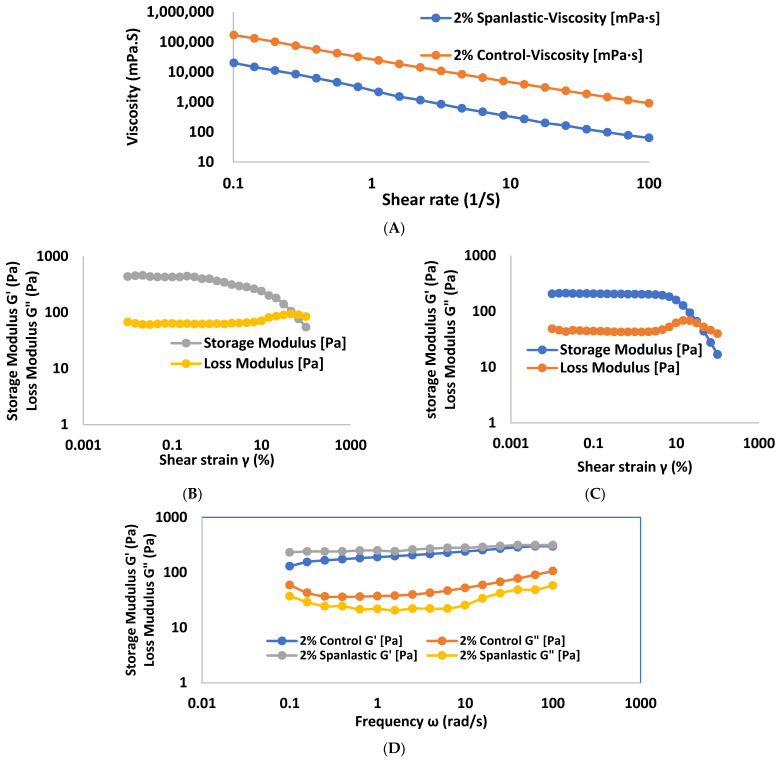
The flow curves (**A**) R-1 gel and C-1 gels determined at 32 °C. (**B**) The amplitude sweeps for R-1 gel. (**C**) The amplitude sweeps for C-1 gel. (**D**) The frequency sweeps for R-1 and C-1 gels. Data are presented as mean ± SD (n = 3).

**Figure 7 pharmaceutics-17-00048-f007:**
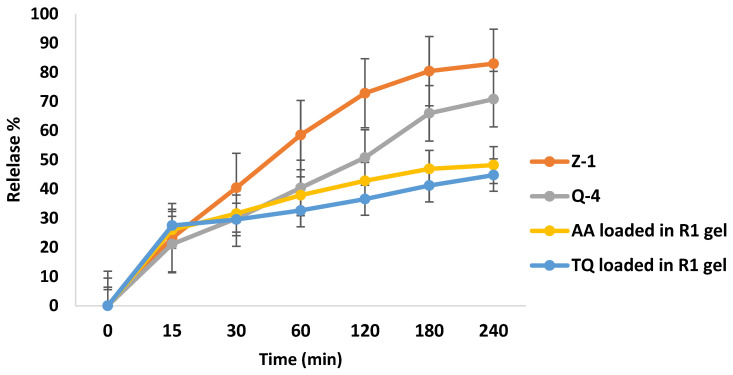
The release profile of Z-1, Q-4, TQ, and AA in R-1 gel. Data are presented as mean ± SD (n = 3).

**Figure 8 pharmaceutics-17-00048-f008:**
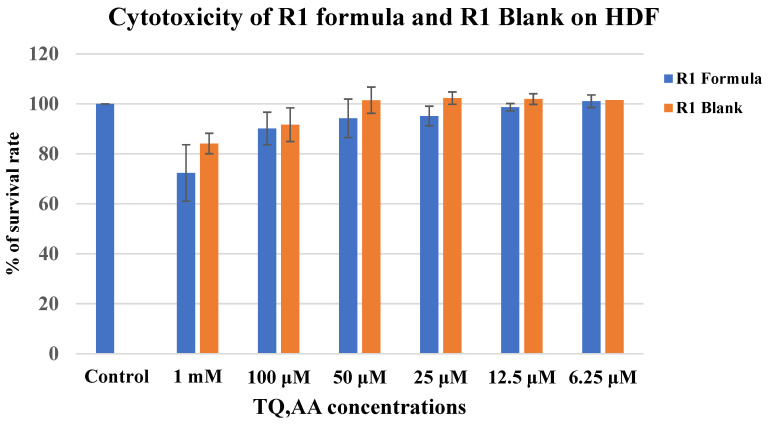
The Effect of R-1 gel and R-1 blank gel on HDF cell line viability compared to control, where no drug is added. Data are presented as mean ± SD (n = 3).

**Figure 9 pharmaceutics-17-00048-f009:**
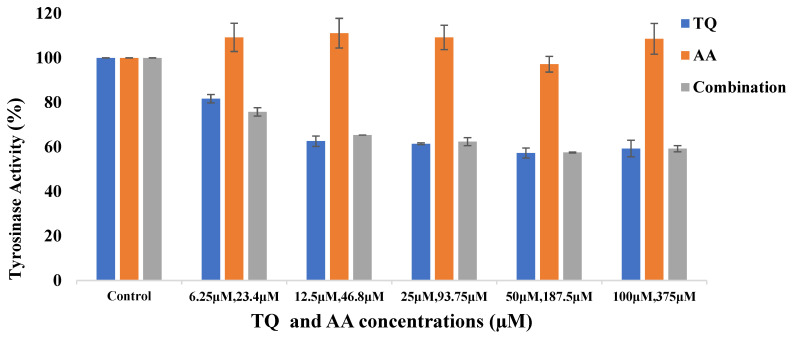
The Tyrosinase Activity of TQ Spanlastics (Q-4), AA Spanlastics (Z-1), and their Combination Compared to Control, where no inhibitor is added. Data are presented as mean ± SD (n = 3).

**Table 1 pharmaceutics-17-00048-t001:** The composition of different formulations for TQ and AA-loaded spanlastics.

Codes	Ratio	Span 60 (mg)	EA Type	EA (mg)	TQ (mg)	AA (mg)	HLB ^1^
Q-1	8:2	80	Tween 80	20	20	-	6.7
Q-2	7:3	70	Tween 80	30	20	-	7.7
Q-3	6:4	60	Tween 80	40	20	-	8.7
Q-4	6:4	60	Tween 20	40	20	-	9.5
Z-1	6:4	300	Tween 20	200	-	100	9.5
Z-2	7:3	350	Tween 20	150	-	100	8.3
Z-3	8:2	400	Tween 20	100	-	100	7.1

^1^ Hydrophilic–lipophilic balance (HLB).

**Table 2 pharmaceutics-17-00048-t002:** Equilibrium solubility of TQ in PBS (pH = 7.4), PBS with (40% PEG-400), and PBS (40% ethanol) at 37 ± 1 °C (Mean ± SD, n = 3).

Solvent	Saturation Solubility (mg/mL)
PBS	0.537 ± 0.124
PBS containing 40% PEG-400	2.228 ± 0.459
PBS containing 40% ethanol	13.450 ± 0.783

**Table 3 pharmaceutics-17-00048-t003:** Results of different characterization parameters for TQ and AA-loaded spanlastics; PS, PDI, ZP, and EE% (mean ± SD, n = 3).

Code	PS (nm)	PDI	ZP	EE%
Q-1	118.90 ± 3.30	0.18 ± 0.01	−15.90 ± 1.91	32.59 ± 1.91
Q-2	113.80 ± 11.00	0.31 ± 0.01	−16.20 ± 0.62	35.72 ± 1.51
Q-3	108.10 ± 0.20	0.31 ± 0.00	−19.00 ± 1.37	41.61 ± 1.48
Q-4	223.40 ± 3.50	0.25 ± 0.00	−21.50 ± 1.72	97.18 ± 2.02
Z-1	133.00 ± 2.80	0.28 ± 0.00	−19.50 ± 1.27	93.08 ± 1.95
Z-2	115.30 ± 1.30	0.18 ± 0.07	−9.40 ± 0.41	83.47 ± 4.99
Z-3	108.60 ± 0.80	0.02 ± 0.02	−8.98 ± 0.07	49.30 ± 0.42

**Table 4 pharmaceutics-17-00048-t004:** The % drug content of TQ and AA from R-1 and C-1. Data are presented as mean ± SD (n = 3).

Gel Code	Recovered Drug	% Recovery
	TQ	95.34 ± 1.86
R-1	AA	99.12 ± 0.62
	TQ	54.57 ± 0.49
C-1	AA	60.72 ± 0.34

**Table 5 pharmaceutics-17-00048-t005:** The LVE Region and Critical Strain (γc) of R-1 and C-1 Gels Determined from the Amplitude-Sweep Experiments.

Gel Code	LVE Region	% (γc)
R-1	0.01–1.00	1.00
C-1	0.01–6.81	6.81

**Table 6 pharmaceutics-17-00048-t006:** Short-term stability study results for Q-4 and Z-1 at 4 °C. PS, PDI, and EE% were studied as stability parameters. Data are represented by mean ± SD (n = 3).

Formulation	Time	PS	PDI	EE%
Q-4	1 month	227.0 ± 2.4	0.20 ± 0.00	98.13 ± 1.21
	2 months	233.4 ± 2.7	0.27 ± 0.01	96.82 ± 1.56
Z-1	1 month	134.4 ± 2.3	0.25 ± 0.00	90.87 ± 5.18
	2 months	136.6 ± 2.6	0.19 ± 0.01	94.66 ± 2.55

**Table 7 pharmaceutics-17-00048-t007:** Short-term stability study for spanlastics gel after 1 month at 4 °C.

Code	Color	Appearance	pH	Spreadability (cm)	%Recovery
R-1	White	Homogeneous	5.50 ± 0.05	3.5 cm	(TQ) 97.45 ± 1.70 (AA) 99.87 ± 1.24

**Table 8 pharmaceutics-17-00048-t008:** Q of TQ and AA permeated across the skin within 5 h, and the amount of TQ and AA deposited in the skin (μg/cm^2^). Data are presented as mean ± SD (n = 3).

Drug	Q (μg/cm^2^)	% Drug Deposition
TQ	65.49 ± 2.01	73.00 ± 0.58
AA	128.75 ± 0.92	74.37 ± 5.65

## Data Availability

The data presented in this study are available on request from the corresponding author. The data are not publicly available due to privacy.

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
