# Peer review of "Fabrication of Thymoquinone and Ascorbic Acid-Loaded Spanlastics Gel for Hyperpigmentation: In Vitro Release, Cytotoxicity, and Skin Permeation Studies"

_pharmaceutics, 2025, doi:10.3390/pharmaceutics17010048_

Round 1
Reviewer 1 Report
Comments and Suggestions for Authors
While the manuscript initially appeared promising, my enthusiasm decreased upon a deeper review due to several critical issues. First, it is poorly written, with numerous grammatical errors and a lack of fluidity, making it difficult to follow. The writing would greatly benefit from revision to improve clarity and readability.
Secondly, there is a significant discrepancy between the study’s stated cosmetic and dermatological purposes. This dual focus must be clearer and more coherent with the research objectives. The authors should clearly define whether the primary goal is pharmaceutical (targeting dermal layers) or cosmetic (remaining in the epidermis) and align the discussion and methodology accordingly.
Finally, some statistical analyses still need to be included, and the presentation of results could be clearer. For instance, key comparisons lack p-values, and the figures and tables could better highlight the differences between formulations.
1. Reference Style
- The references appear inconsistently formatted and must adhere to the journal’s required style.
2. Grammar and Sentence Clarity
The manuscript contains grammatical errors, awkward phrasing, and inconsistencies that hinder readability. Proofreading by a native English speaker or professional editing service is strongly recommended to ensure the manuscript meets academic language standards.
- Line 39: Subject-verb agreement issues are present
- Lines 66-67: The manuscript mentions bioavailability and routes of administration for ascorbic acid (AA) but transitions awkwardly. Rewrite to clarify the challenges of dermal delivery.
- Line 69: The verb is missing
- Line 322: The verb is missing
- Line 391: The verb is missing
3. Introduction
- Line 59-60: Include the log P and solubility values for TQ, which are critical for evaluating its formulation challenges.
- Line 60: The chemical instability of TQ is noted but not described. Specify if it's due to oxidation, hydrolysis, or photodegradation.
- Lines 69-77: The authors describe the benefits of dermal drug delivery. Dermal drug delivery suggests pharmaceutical applications, while cosmetics target the epidermis. Clarify whether the study focuses on dermatological or cosmetic use.
- Line 78: Expand on the structure, composition, and advantages of spanlastic vesicles to provide a complete understanding.
- Lines 84-90: The objectives could be clearer. Rephrase to explicitly state the purpose of encapsulating TQ and AA, improving their stability and permeation.
4. Methods Section
- Line 100: Introduce RP-HPLC before listing its parameters.
- Line 101: Replace “Agilant” with “Agilent.”
- Line 108: Include retention times for TQ and AA in HPLC analysis.
- Line 112: PEG 400 is used but needs to be listed in the materials section.
- Line 120: Resolve the inconsistency between "ether injection" (line 86) and "ethanol injection" (line 120).
- Line 122: Define "mixed well" with a precise scientific method (e.g., duration, mixing method).
- Line 124: Specify the edge activator used and add a reference to Table 1.
- Line 138: Clarify “HLB” in Table 1.
- Line 146 and 153: Explain why sonication was used for particle size and zeta potential analysis, as it could alter vesicle properties.
- Line 159: I have some concerns about vesicle integrity during ultracentrifugation. Could this method disrupt vesicles and release encapsulated drugs?
- Line 250: Rephrase "into the rat skin" to "onto the rat skin."
5. Results and Discussion
- Line 331, 570, 572: Ensure tables adhere to journal guidelines.
- Lines 338-340: Move the comparison of spanlastics and niosomes to the introduction to justify the choice of spanlastics. Expand on the advantages, particularly regarding drug profiles and delivery routes. Provide a detailed explanation of why spanlastics outperform niosomes regarding encapsulation efficiency, release profiles, and permeability.
- Line 344-345: Clarify how the TEM analysis supports drug location in vesicles. This claim requires supporting data.
- Lines 349-351: Contrary to the authors' claims, there is a size discrepancy between TEM and DLS results that should be addressed statistically or with standard deviations.
- Line 361-362: Reassess the cosmetic application claim, given the vesicles' dermal penetration.
- Line 460: Explain how gel colour change indicates drug loading.
- Line 466–469: The method for measuring the gel spreadability index is missing. Add this to Section 2.7.2.
- Line 535: Illustrate the release profiles of spanlastics and spanlastic gels into a single figure and section to allow for direct comparison. Discuss how the gel matrix affects the release rates.
- Line 573: Add figures to show permeation profiles and include formulation names in the text (C1 and R1).
- Lines 601-610: Include statistical significance for cell viability data in the figure or discussion.
- Lines 618-631: Report statistical significance for the concentration-dependent effects in tyrosinase inhibition.
6. Abstract and conclusions
The abstract provides a good overview of the study but needs more specificity in key results, such as statistical significance and quantitative comparisons. Additionally, the dual focus on pharmaceutical and cosmetic applications creates ambiguity. The authors are encouraged to clarify the scope of the study, highlight critical findings (e.g., encapsulation efficiency, tyrosinase inhibition), and rephrase for more significant impact.
The conclusion effectively summarises the findings but would benefit from more specificity and acknowledgement of limitations. To enhance the manuscript's relevance and impact, it is recommended that the authors explicitly define the intended application (pharmaceutical or cosmetic), address study limitations (e.g., reliance on ex vivo data), and propose future directions.
Comments on the Quality of English Languageit is poorly written, with numerous grammatical errors and a lack of fluidity, making it difficult to follow. The writing would greatly benefit from revision to improve clarity and readability.
Author Response
Response to reviewers
We would like to thank the reviewers for the careful and thorough reading of this manuscript and for the thoughtful comments and constructive suggestions, which help to improve the quality of this manuscript. Kindly find our response to reviewers. We have been able to incorporate changes to reflect most of the suggestions provided by the reviewers. We have tracked the changes within the manuscript. If there are any questions or it is not clear, please do let me know.
Here is a point-by-point response to the reviewers’ comments and concerns
Reviewer 1
- Reference Style
- The references appear inconsistently formatted and must adhere to the journal’s required style.
Thank you for your comment. We revised the references and adhered to the journal style.
- Grammar and Sentence Clarity
The manuscript contains grammatical errors, awkward phrasing, and inconsistencies that hinder readability. Proofreading by a native English speaker or professional editing service is strongly recommended to ensure the manuscript meets academic language standards.
- Line 39: Subject-verb agreement issues are present
- Lines 66-67: The manuscript mentions bioavailability and routes of administration for ascorbic acid (AA) but transitions awkwardly. Rewrite to clarify the challenges of dermal delivery.
- Line 69: The verb is missing
- Line 322: The verb is missing
- Line 391: The verb is missing
Thank you for your comment. We appreciate your observation regarding the language and readability of the manuscript. We will thoroughly revise the text to address the grammatical errors, awkward phrasing, and inconsistencies. Additionally, we sought assistance from a professional editing service to ensure the manuscript meets the highest academic language standards.
- Introduction
- Line 59-60: Include the log P and solubility values for TQ, which are critical for evaluating its formulation challenges.
Line 59-60 : Thank you for your comment. We have included the log P and solubility values for TQ and the changes visible in the revised manuscript.
- Line 60: The chemical instability of TQ is noted but not described. Specify if it's due to oxidation, hydrolysis, or photodegradation.
Line 60 : Thank you for your comment. We have described the chemical instability of TQ and the changes visible in the revised manuscript.
- Lines 69-77: The authors describe the benefits of dermal drug delivery. Dermal drug delivery suggests pharmaceutical applications, while cosmetics target the epidermis. Clarify whether the study focuses on dermatological or cosmetic use.
Line 80: Thank you for your comment. The present study addresses hyperpigmentation as clarified in the title, focusing on delivering TQ and AA to the skin layers where melanogenesis occurs. Melanocytes are present in the epidermis and hair follicles. Therefore, the goal of the study is to achieve dermal delivery that can effectively reduce melanin production. This approach avoids systemic side effects, offering a promising strategy for treating hyperpigmentation in a controlled manner.
- Line 78: Expand on the structure, composition, and advantages of spanlastic vesicles to provide a complete understanding.
Line 85: Thank you for your comment. We have described the structure, composition, and advantages of spanlastic vesicles to provide a complete understanding and the changes visible in the revised manuscript.
- Lines 84-90: The objectives could be clearer. Rephrase to explicitly state the purpose of encapsulating TQ and AA, improving their stability and permeation.
Line 98: Thank you for your comment. We have revised and rewritten the purpose of the study and the changes visible in the revised manuscript.
- Methods Section
- Line 100: Introduce RP-HPLC before listing its parameters.
- Line 101: Replace “Agilant” with “Agilent.”
- Line 108: Include retention times for TQ and AA in HPLC analysis.
- Line 112: PEG 400 is used but needs to be listed in the materials section.
- Line 120: Resolve the inconsistency between "ether injection" (line 86) and "ethanol injection" (line 120).
- Line 122: Define "mixed well" with a precise scientific method (e.g., duration, mixing method).
- Line 124: Specify the edge activator used and add a reference to Table 1.
- Line 138: Clarify “HLB” in Table 1.
- Line 146 and 153: Explain why sonication was used for particle size and zeta potential analysis, as it could alter vesicle properties.
- Line 159: I have some concerns about vesicle integrity during ultracentrifugation. Could this method disrupt vesicles and release encapsulated drugs?
- Line 250:Rephrase "into the rat skin" to "onto the rat skin."
Thank you for your comment. We have addressed all the comments and the changes visible in the revised manuscript
- Results and Discussion
- Line 331, 570, 572: Ensure tables adhere to journal guidelines.
Thank you for your comment. We have revised them and adhered to journal guidelines and the changes visible in the revised manuscript
- Lines 338-340: Move the comparison of spanlastics and niosomes to the introduction to justify the choice of spanlastics. Expand on the advantages, particularly regarding drug profiles and delivery routes. Provide a detailed explanation of why spanlastics outperform niosomes regarding encapsulation efficiency, release profiles, and permeability.
Thank you for your comment. We have moved it to the introduction section and the changes visible in the revised manuscript
- Line 344-345: Clarify how the TEM analysis supports drug location in vesicles. This claim requires supporting data.
Line 363 : Thank you for your comment. Upon review, we acknowledge that the TEM analysis alone does not provide sufficient evidence to confirm the exact drug location within the vesicles. But in combination with X-ray energy dispersive spectroscopy EDX, TEM can provide nanoscale elemental analysis that is fundamental to properly localizing the drug inside the lipid vectors. Therefore, we have deleted the sentence to ensure accuracy and avoid unsupported claims and the changes visible in the revised manuscript
- Lines 349-351: Contrary to the authors' claims, there is a size discrepancy between TEM and DLS results that should be addressed statistically or with standard deviations.
Line 367 : Thank you for your comment. To address this, we included the standard deviations for both methods.
- Line 361-362: Reassess the cosmetic application claim, given the vesicles' dermal penetration.
Line 383 : Thank you for your comment. We have revised the sentence and added dermatological application instead of cosmeceutical purposes and the changes visible in the revised manuscript.
- Line 460:Explain how gel colour change indicates drug loading.
Line 383 : Thank you for your comment. We have clarified the sentence and the changes visible in the revised manuscript. The color change from yellow to white indicates successful drug loading and encapsulation of TQ within the spanlastic gel. Free TQ is yellow, but upon entrapment, its color is masked within the vesicle and gel matrix. High drug recovery percentages further confirm the successful encapsulation.
- Line 466–469:The method for measuring the gel spreadability index is missing. Add this to Section 2.7.2.
Thank you for your comment. We have revised the sentence and added the gel spreadability index and the changes visible in the revised manuscript.
- Line 535:Illustrate the release profiles of spanlastics and spanlastic gels into a single figure and section to allow for direct comparison. Discuss how the gel matrix affects the release rates.
Thank you for your comment. We have added the release profiles of spanlastics and spanlastic gels as a single figure and the changes visible in the revised manuscript.
- Line 573: Add figures to show permeation profiles and include formulation names in the text (C1 and R1).
Thank you for your comment. We have chosen to represent the results of permeation studies in a table to clearly show the amount deposited in the skin layers, as our focus is on dermal delivery rather than transdermal delivery. This approach allows for a more precise understanding of how the formulation interacts with and is retained in the skin.
- Lines 601-610: Include statistical significance for cell viability data in the figure or discussion.
- Lines 618-631: Report statistical significance for the concentration-dependent effects in tyrosinase inhibition.
Thank you for your comment. We have included in the discussion and the changes visible in the revised manuscript.
- 6. Abstract and conclusions
The abstract provides a good overview of the study but needs more specificity in key results, such as statistical significance and quantitative comparisons. Additionally, the dual focus on pharmaceutical and cosmetic applications creates ambiguity. The authors are encouraged to clarify the scope of the study, highlight critical findings (e.g., encapsulation efficiency, tyrosinase inhibition), and rephrase for more significant impact.
The conclusion effectively summarises the findings but would benefit from more specificity and acknowledgement of limitations. To enhance the manuscript's relevance and impact, it is recommended that the authors explicitly define the intended application (pharmaceutical or cosmetic), address study limitations (e.g., reliance on ex vivo data), and propose future directions.
Thank you for your comment. We have revised the abstract and conclusion sections and the changes visible in the revised manuscript.
Reviewer 2 Report
Comments and Suggestions for Authors
Ms.Nr: pharmaceutics-3352287
Zaid Alkilani et al:”Fabrication of Thymoquinone and Ascorbic Acid-Loaded Spanlastics Gel for hyperpigmentation: In vitro release, cytotoxicity and skin permeation studies””
This is a comprehensive and interesting study aiming the development and wide characterization of a spanlastics gel loaded with natural agents, TQ and AA, for dermal delivery for the treatment of hyperpigmentation. The successfully designed R-1 product as nanovesicle has several advantages like less toxicity, good rheology and better stability than the conventional C-1 gel. The special merit of the work is the multilateral investigation of the new formulations including rheological properties, short time stability, in vitro release and permeability. In addition, the mechanism of action and the toxicity were also studied. The topic is current, the novelty of the work is significant, and the scientific soundness of the work is meaningful.
The manuscript is logically constructed and clear. The Abstract is enough informative, the Introduction provides sufficient insight to the usage of TQ, its mechanism of action, and safety as lightening agent. The cited references are relevant and mostly from the last 5-8 years. However, the text is required a thorough editing control. I suggest the acceptance for publication after minor revision. See remarks below.
1. In the preparation process 60-65 °C was applied. How it does impact the stability of the APIs, knowing both as temperature and light sensitive agents. Authors should address this issue.
2. Use the correct physicochemical term equilibrium (or thermodynamic) solubility instead of saturation solubility.
3. Use SI units everywhere in the text (not ml, gm, etc.)
4. The numbering of tables from Table 3 is erroneous, correct all over in the text.
5. Smaller mistakes must be corrected at the indicated row numbers:
86: “ether inection” to ethanol injection
169: first the sentence is not completed
209: it is rather, water : n-propanol 20:80 mixture than, as written
250: give the volume of the receiver solvent
342: correct the figure legend (Z-1 contains AA and Q-4 TQ)
419: harmonize the numbers pointing to the peak of blank spanlastic: in the text 3310 cm-1, while on Figure 4. A and B 3330 and 3332 cm-1
468: on Figure 6, the spot on B panel seems to be double of that on C panel, while in the text only small difference 3.65 vs 3.03 cm is given. Why?
522: Gel code is C-1 not Z-3
Author Response
Response to reviewers
We would like to thank the reviewers for the careful and thorough reading of this manuscript and for the thoughtful comments and constructive suggestions, which help to improve the quality of this manuscript. Kindly find our response to reviewers. We have been able to incorporate changes to reflect most of the suggestions provided by the reviewers. We have tracked the changes within the manuscript. If there are any questions or it is not clear, please do let me know.
Here is a point-by-point response to the reviewers’ comments and concerns
1.In the preparation process 60-65 °C was applied. How it does impact the stability of the APIs, knowing both as temperature and light sensitive agents. Authors should address this issue.
The temperature of 60–65 °C was applied for a short duration during spanlastics preparation. Light-sensitive compounds were protected by conducting the process using amber containers and wrapping it with aluminum foils. Stability of both was confirmed by the high drug recovery (TQ: 95.34 ± 1.86%, AA: 99.12 ± 0.62%).
- Use the correct physicochemical term equilibrium (or thermodynamic) solubility instead of saturation solubility.
- Use SI units everywhere in the text (not ml, gm, etc.)
- The numbering of tables from Table 3 is erroneous, correct all over in the text.
- Smaller mistakes must be corrected at the indicated row numbers:
86: “ether inection” to ethanol injection
169: first the sentence is not completed
209: it is rather, water : n-propanol 20:80 mixture than, as written
250: give the volume of the receiver solvent
342: correct the figure legend (Z-1 contains AA and Q-4 TQ)
419: harmonize the numbers pointing to the peak of blank spanlastic: in the text 3310 cm-1, while on Figure 4. A and B 3330 and 3332 cm-1
522: Gel code is C-1 not Z-3
Thank you for your comment. We have addressed all comments and the changes visible in the revised manuscript.
468: on Figure 6, the spot on B panel seems to be double of that on C panel, while in the text only small difference 3.65 vs 3.03 cm is given. Why?
Thank you for your feedback. Upon further consideration, we agree that the discrepancy between the spot sizes on panels B and C in Figure 6 could be misleading due to scaling issues during image capture. These images were captured using a digital camera, which may have introduced minor distortions or scaling effects. To ensure clarity and accuracy, we have decided to remove this image. Thank you for bringing this to our attention.

Reviewer 3 Report
Comments and Suggestions for Authors
In this article, the authors developed Thymoquinone (TQ) and Ascorbic Acid (AA)-Loaded Spanlastics Gel for the treatment of hyperpigmentation. The as developed nanovesicles and nanovesicle loaded hydrogel were well characterized using different techniques. The authors also performed ex vivo studies to examine the drug diffusion into the skin. This is a novel and interesting study. However, the following comments/suggestions need to address before publication.
In Ex vivo drug permeation studies, controls i.e. unloaded TQ and AA, TQ and AA loaded nanovesicles (without gel) need to be included and compared.
In Tyrosinase inhibition activity also need to be compared between free TQ and AA, blank nanovesicles (without TQ and AA), nanovesicles loaded with TQ and AA, nanovesicles loaded gel.
This manuscript has some grammatical mistakes and typos. So, please have a thorough look into it.
Author Response
We would like to thank the reviewers for the careful and thorough reading of this manuscript and for the thoughtful comments and constructive suggestions, which help to improve the quality of this manuscript. Kindly find our response to reviewers. We have been able to incorporate changes to reflect most of the suggestions provided by the reviewers. We have tracked the changes within the manuscript. If there are any questions or it is not clear, please do let me know.
Here is a point-by-point response to the reviewers’ comments and concerns
In Ex vivo drug permeation studies, controls i.e. unloaded TQ and AA, TQ and AA loaded nanovesicles (without gel) need to be included and compared.
Thank you for your comment. While we have conducted in vitro release studies for TQ and AA vesicles to understand the release profile. For permeation studies, it is essential to assess the performance of the final dosage form. This allows us to evaluate the formulation as a whole, considering the impact of all components, including the gel matrix, on drug permeation.
In Tyrosinase inhibition activity also need to be compared between free TQ and AA, blank nanovesicles (without TQ and AA), nanovesicles loaded with TQ and AA, nanovesicles loaded gel.
Thank you for your valuable suggestion regarding comparing tyrosinase inhibition activity across free TQ, AA, blank nanovesicles, nanovesicles loaded with TQ and AA, and nanovesicle-loaded gel formulations. While we acknowledge the importance of such comprehensive comparisons, the focus of our current study was to evaluate the efficacy of the optimized spanlastic gel formulation encapsulating TQ and AA.
The inhibitory effects of free TQ on tyrosinase activity are well-documented. For instance, Jeong et al. (2020) demonstrated that TQ significantly inhibits melanogenesis by downregulating tyrosinase expression in B16F10 mouse melanoma cells (https://www.spandidos-publications.com/ijo/56/1/379). Similarly, the role of AA in modulating tyrosinase activity has been explored. A study by Wen et al. (2021) reported that ascorbic acid exhibits tyrosinase inhibitory activity (https://pubmed.ncbi.nlm.nih.gov/34730855/).
It is also worth noting that the stability of free TQ and AA is a significant challenge, as these compounds are prone to degradation under light, heat, and oxidative conditions. Previous studies have highlighted the poor aqueous solubility and chemical instability of TQ.
Encapsulation within nanovesicles enhances the stability of these compounds by providing a protective matrix, shielding them from external degradative factors. Our findings demonstrated improved drug recovery rates and prolonged release profiles for TQ and AA when encapsulated in spanlastic nanovesicles and incorporated into gel formulations. This ensures greater bioavailability and efficacy compared to their free forms.
While we appreciate the merit of including comparative analyses with free compounds and blank formulations, such evaluations are beyond the scope of this manuscript. We focused on developing and characterizing the novel nanovesicle-loaded gel system as a means to enhance the stability and efficacy of TQ and AA for dermal delivery. Future research could build upon this work to explore additional comparative analyses.
We hope this explanation clarifies the rationale for our study design, and we appreciate your insightful feedback. Please let us know if you require any further information.
This manuscript has some grammatical mistakes and typos. So, please have a thorough look into it.
Thank you for your feedback. I thoroughly reviewed the manuscript to address any grammatical mistakes and typos.

Round 2
Reviewer 3 Report
Comments and Suggestions for Authors
The authors addressed the concerns of reviewer. So, it can be accepted for publication.